# The Neuroprotective Role of Coenzyme Q10 Against Lead Acetate-Induced Neurotoxicity Is Mediated by Antioxidant, Anti-Inflammatory and Anti-Apoptotic Activities

**DOI:** 10.3390/ijerph16162895

**Published:** 2019-08-13

**Authors:** Al Omar S. Yousef, Alkhuriji A. Fahad, Ahmed E. Abdel Moneim, Dina M. Metwally, Manal F. El-khadragy, Rami B. Kassab

**Affiliations:** 1Zoology Department, Faculty of Science, King Saud University, Riyadh 11451, Saudi Arabia; 2Zoology and Entomology Department, Faculty of Science, Helwan University, Cairo 11795, Egypt; 3Parasitology Department, Faculty of Veterinary Medicine, Zagazig University, Zagazig 12878, Egypt

**Keywords:** lead, coenzyme Q10, Nrf2/HO-1 pathway, inflammation, neurotransmission, apoptosis, brain

## Abstract

Heavy metal exposure, in lead (Pb) particularly, is associated with severe neuronal impairment though oxidative stress mediated by reactive oxygen species, and antioxidants may be used to abolish these adverse effects. This study investigated the potential neuroprotective role of coenzyme Q10 (CoQ_10_) against lead acetate (PbAc)-induced neurotoxicity. Twenty-eight male Wistar albino rats were divided into four equal groups (*n* = 7) and treated as follows: the control group was injected with physiological saline (0.9% NaCl); the CoQ_10_ group was injected with CoQ_10_ (10 mg/kg); PbAc group was injected with PbAc (20 mg/kg); PbAc + CoQ_10_ group was injected first with PbAc, and after 1 h with CoQ_10_. All groups were injected intraperitoneally for seven days. PbAc significantly increased cortical lipid peroxidation, nitrate/nitrite levels, and inducible nitric oxide synthase expression, and decreased glutathione content, superoxide dismutase, catalase, glutathione peroxidase, glutathione reductase activity and mRNA expression, as well as nuclear factor erythroid 2–related factor 2 (Nrf2) and homoxygenase-1 (HO-1) expression. PbAc also promoted the secretion of interleukin-1ß and tumor necrosis factor-α, inhibited interleukin-10 production, triggered the activation of pro-apoptotic proteins, and suppressed anti-apoptotic proteins. Additionally, PbAc increased the cortical levels of serotonin, dopamine, norepinephrine, GABA, and glutamate, and decreased the level of ATP. However, treatment with CoQ_10_ rescued cortical neurons from PbAc-induced neurotoxicity by restoring the balance between oxidants and antioxidants, activating the Nrf2/HO-1 pathway, suppressing inflammation, inhibiting the apoptotic cascade, and modulating cortical neurotransmission and energy metabolism. Altogether, our findings indicate that CoQ_10_ has beneficial effects against PbAc-induced neuronal damage through its antioxidant, anti-inflammatory, anti-apoptotic, and neuromodulatory activities.

## 1. Introduction

Exposure to heavy metals is a major health concern worldwide due to their long biological half-lives and irreversible effects on different body systems [1]. Lead (Pb), a naturally-distributed heavy metal found mainly in combination with sulfur, is characterized by poor conductivity, malleability, high density, and resistance to corrosion. These physio-chemical properties have meant that Pb is widely used in a variety of industries. Humans are mainly exposed to Pb through the ingestion and inhalation of contaminated sources such as water, food, soil, and air [2]. Once absorbed, Pb diffuses into the soft tissues, causing adverse effects in the kidney, hematopoietic, reproductive, and cardiovascular systems, with the central nervous system identified as one of the most affected organs following Pb exposure [3,4]. Children are the most susceptible to Pb toxicity and are at risk of several irreversible impairments including neurodevelopmental defects, intellectual disabilities, and behavioral deficits [5]. Although the precise mechanisms involved in Pb-induced neurotoxicity remain unclear, several factors have been suggested to play a key role, including oxidative stress, neuroinflammation, apoptosis, and alterations to neurochemical, molecular, and cellular signaling.

The use of antioxidants to prevent or delay the adverse reactions produced by heavy metals has received great interest among researchers. Coenzyme Q_10_ (CoQ_10_), a benzoquinone compound widely known as ubiquinone, is ubiquitous in plants, animals, and bodily tissues as a lipid soluble agent. Large amounts of CoQ_10_ are found in the muscles, liver, kidney, and heart due to their high energy requirements [6]. It is mainly biosynthesized from mevalonate, tyrosine, and vitamins C, B2, B9, and B12 in the Golgi apparatus and mitochondria, then transported into the circulation by low-density lipoproteins [7]. During metabolism and inside the mitochondria, the electron transport chain uses CoQ_10_ as an electron carrier for oxidative phosphorylation and ATP production. Outside the mitochondria, CoQ_10_ acts as a potent lipophilic antioxidant, providing cellular protection against free radical production along with other antioxidants [6]. It has been shown that aging and its related changes, such as oxidative stress, affect the mitochondria and decrease CoQ_10_ content [8]. The therapeutic efficacy of CoQ_10_ has been confirmed in mitochondrial dysfunction- and oxidative challenge-related disorders such as cancer, diabetes, and cardiovascular and neurodegenerative disorders [9]. Its beneficial effects also extend to minimizing heavy metal-induced neuronal impairments in experimental animal models [10,11].

To our knowledge no pre-clinical studies exist relating to the effect of CoQ10 on lead acetate-induced neurotoxicity. Therefore, this study aimed to investigate the potential protective role of CoQ_10_ against Pb-induced neurotoxicity by evaluating the oxidative status, release of pro-inflammatory cytokines, expression of apoptotic proteins, and the levels of ATP, both monoaminergic and aminoacidergic in the frontal cortex of rats. 

## 2. Materials and Methods 

### 2.1. Chemicals

CoQ_10_ (C_59_H_90_O_4_; CAS Number: 303-98-0) and lead(II) acetate trihydrate (Pb(CH_3_CO_2_)_2_·3H_2_O; CAS Number 6080-56-4), were purchased from Sigma-Aldrich (St. Louis, MO, USA). All other chemicals and reagents used in these experiments were analytical grade and double-distilled water was used as a solvent. CoQ_10_ was dissolved in saline solution (0.9% NaCl) containing 1% Tween 80 (*v*:*v*) by stirring overnight at room temperature.

### 2.2. Experimental Animals

Male Wistar albino rats (150–170 g; 10 weeks old) were obtained from VACSERA (Giza, Egypt), and kept in wire polypropylene cages under standard laboratory conditions (12 h light: 12 h dark; 25 ± 2 °C). The rats were supplied with water and food ad libitum and allowed to acclimatize for one week before experiments were started.

### 2.3. Experimental Design

In order to evaluate the potential protective role of CoQ_10_ against lead acetate (PbAc)-induced cortical damage, the rats were divided into four groups of seven, as follows:(1)Control group injected intraperitoneally (i.p.) with 0.1 mL of saline containing 1% Tween 80 (*v*:*v*).(2)CoQ_10_ group injected i.p. daily at a dose of 10 mg/kg bwt, according to Fouad and Jresat [12].(3)PbAc group injected i.p. at a dose of 20 mg/kg bwt. The selection of lead acetate dose was based on our previous studies, as well as literature data [13,14,15], which is equal to 1/30 LD_50_ for rats to induce acute neurotoxicity model [13].(4)PbAc and CoQ_10_ injected group i.p. first with PbAc, then after 1 h injected with CoQ_10_ using the same mentioned doses.

The animals were killed by fast decapitation 24 h after the last injection. The brain was removed carefully and washed twice in ice-cold Tris-HCl (50 mM; pH 7.4), and the frontal cortex was isolated. To determine biochemical parameters, the frontal cortex was homogenized in ice-cold medium containing Tris-HCl (50 mM; pH 7.4) to give a 10% (*w*/*v*) homogenate. For neurochemical determination, cortical tissue was homogenized in 75% aqueous HPLC grade methanol (10% *w*/*v*). The homogenates were spun for 10 min at 3000 × *g* at 4 °C. The total cortical protein content was estimated using the method described by Lowry et al. [16] using bovine serum albumin as a reference standard.

### 2.4. Lead Concentration in the Cortical Tissue

The cortical lead content was determined using the protocol described by [17]. Cortical samples were dried at 60 °C and combusted for 24 h at 150 °C in an oven. Thereafter, samples were added to a hot solution of 1 M HNO_3_, adjusted to 50 mL with deionized water and then analyzed using a flame atomic absorption spectrophotometer (Perkin-Elmer, 3100) at 283.3 nm. The concentration of lead in the cortex is given as µg/g wet tissue weight.

### 2.5. Oxidative Stress Markers in the Cortical Tissue

Cortical lipid peroxidation (LPO) was determined using 1 mL of 10% trichloroacetic acid and 1 mL of 0.67% thiobarbituric acid in a boiling water bath for 30 min. Thiobarbituric acid-reactive substances were assessed by their absorbance at 535 nm and expressed as the amount of malondialdehyde (MDA) formed [18]. The level of nitric oxide in the cortical homogenate was assessed using the optimized acid reduction method in an acidic medium, in the presence of nitrite. This reaction couples nitrous acid diazotized sulfanilamide with N-(1-naphthyl) ethylenediamine, and the resultant bright reddish-purple azo dye can be measured at 540 nm [19]. In addition, the level of cortical glutathione (GSH) was determined by the reduction of 5,5′-Dithiobis(2-nitrobenzoic acid) (Ellman’s reagent) with GSH to form a yellow compound. Protein denaturation was performed before the addition of Ellman’s reagent to avoid interaction with the other thiol proteins. The color is directly proportional to the GSH concentration, and its absorbance was measured at 405 nm [20].

### 2.6. Antioxidant Status of the Cortical Tissue

The activity of cortical superoxide dismutase (SOD) was determined by measuring the ability of SOD to suppress the phenazine methosulfate-mediated reduction of nitroblue tetrazolium (NBT) dye and one unit of SOD activity is given as the enzyme protein amount causing 50% inhibition in NBT reduction rate. Cortical catalase (CAT) activity was estimated by adding 50 µL of cortical sample to H_2_O_2_ (30 mM) in a potassium phosphate buffer (50 mM; pH 8.0). H_2_O_2_ decomposition was measured at 340 nm at 20 s intervals for 2 min, and catalase activity was reported in units. Glutathione reductase (GR) activity was determined indirectly by its catalysis of glutathione reduction in the presence of NADPH, which is oxidized to NADP^+^ and is accompanied by a decrease in absorbance at 340 nm, the enzyme activity was calculated using molar extinction coefficient (6.22 × 10^3^M^−1^cm^−1^). Cortical glutathione peroxidase (GPx) activity was assessed using the protocol described by Paglia and Valentine [21], which indirectly measures the activity of GPx. Oxidized glutathione, produced by the reduction of organic peroxide by GPx, is recycled to its reduced state by the enzyme glutathione reductase, again decreasing the absorbance at 340 nm, which can then be measured and the enzyme activity was calculated using molar extinction coefficient (6.22 × 10^3^M^−1^cm^−1^).

### 2.7. Inflammation Marker Assays

Cortical levels of tumor necrosis factor-α (TNF-α; Catalog # BMS622 with an analytical sensitivity of 11.0 pg/mL), interleukin-1β (IL-1β; Catalog # BMS630 with an analytical sensitivity of 4.0 pg/mL), and interleukin-10 (IL-10; Catalog # BMS629 with an analytical sensitivity of 1.5 pg/mL) were assayed using kits purchased from ThermoFisher Scientific (Waltham, MA, USA) according to the manufacturer’s protocols.

### 2.8. Quantitative Real Time PCR

Total RNA was isolated from the cortical tissue using an RNeasy Plus Minikit (Qiagen, Valencia, CA, USA). cDNA was prepared in 20 µl reaction volume using the RevertAid™ H Minus Reverse Transcriptase (Fermentas, Thermo Fisher Scientific Inc., Burlington, Canada) and real-time PCR analysis run in triplicate using Power SYBR® Green (Life Technologies, Carlsbad, CA, USA) on an Applied Biosystems 7500 Real-Time PCR System. The PCR reaction thermal program was 95 °C for 4 min, 40 cycles at 95 °C for 10 s, 60 °C for 30 s and 72 °C for 10 s. detecting the ^delta^ Ct. Relative gene expression values were normalized to β-actin based on a preliminary study. Primer sequences are provided in Table 1 as previously published [22]. All the qRT-PCR experiments and data analysis in the present research were performed in accordance with the MIQE guidelines [23].

### 2.9. Histological Changes

The cerebral cortex was isolated and fixed in 10% neutral buffered formalin for 24 h, dehydrated in ethyl alcohol, cleared in xylene, and mounted in molten paraplast. The cerebral cortex blocks were then divided into 4–5 µm sections, stained with hematoxylin-eosin, and examined using a Nikon (Eclipse E200-LED, Tokyo, Japan) microscope.

### 2.10. Apoptotic Proteins Quantification

The levels of the pro-apoptotic proteins (Bax (BioVision, Inc. (San Francisco, CA, USA) Catalog # E4513) and caspase-3 (Sigma-Aldrich (Saint-Louis, MO, USA) Catalog # CASP3C-1KT)) and the anti-apoptotic protein (Bcl-2, Cusabio (Wuhan, China) Catalog # CSB-E08854r) in the cortical homogenate were estimated using commercial kits according to the manufacturer’s protocols.

### 2.11. Neurochemical Changes in the Cortical Tissue

The monoamine content of the cortical tissue was estimated using the method described by Pagel et al. [24]. GABA and glutamate content were measured according the precolumn PITC derivatization technique described by Heinrikson and Meredith [25], whilst ATP levels were assessed using the method of Teerlink et al. [26].

### 2.12. Ethic Statement

All experimental procedures were performed according to the European Community Directive (86/609/EEC) national rules on animal care, carried out in accordance with the NIH Guidelines for the Care and Use of Laboratory Animals 8th edition, and approved by the Institutional Animal Ethics Committee guidelines for animal care and use at Helwan University.

### 2.13. Statistical Analysis

Data are presented as the mean ± standard deviation (SD). For multiple variable comparisons, data were analyzed by one-way analysis of variance (ANOVA). Duncan’s test was used as a post hoc test to compare significance between groups, and *P* ˂ 0.05 was considered significant.

## 3. Results

### 3.1. Pb Concentration in the Cortical Tissue

The Pb concentrations in the cortical homogenates were estimated at the end of the experiments, using an atomic absorption spectrophotometer. Pb content was significantly increased (*P* < 0.05) in the cerebral cortex of rats injected with 20 mg/kg PbAc for 5 consecutive days compared with those of control rats. In addition, there was no significant change in the Pb concentration in the cortical homogenates of CoQ_10_ injected rats compared with those of control rats. Interestingly, the cortical homogenates of rats treated with CoQ_10_ showed a significant decrease (*P* < 0.05) in Pb level compared to those of the PbAc injected group (Figure 1).

### 3.2. PbAc-Induced Oxidative Damage in the Cortical Tissue

To evaluate the oxidative damage in cortical tissue following PbAc-exposure and the potential protective role of CoQ_10_, the levels of major oxidants and antioxidants were measured. The cortical homogenate of rats intraperitoneally injected with PbAc exhibited a significant (*P* < 0.05) increase in LPO, nitrate/nitrite (nitric oxide, NO) levels, and iNOS expression, and a significant decrease in GSH content, SOD, CAT, GPx, and GR activity compared with the control rats. No significant changes in the oxidative parameters were observed in the CoQ_10_ treated rats compared with the control rats. On the other hand, CoQ_10_ significantly (*P* < 0.05) attenuated oxidative damage following PbAc exposure by restoring the balance between pro-oxidants (LPO, nitric oxide, and iNOS expression) and enzymatic and non-enzymatic antioxidants (GSH, SOD, CAT, GPx, and GR) in the cortical tissue (Figure 2 and Figure 3).

Consistent with the biochemical data, RT-qPCR revealed a significant (*P* < 0.05) decrease in the cortical mRNA expression of SOD2, CAT, and GPx1 in PbAc-injected rats compared with the control rats. However, there was no significant change in the expression of these genes in rats injected with 10 mg/kg CoQ_10_ for 5 days compared with the control rats. Furthermore, the injection of CoQ_10_ 1 h after PbAc significantly downregulated cortical antioxidant mRNA expression compared with the rats exposed to PbAc (Figure 3).

To further determine the molecular mechanisms underlying CoQ_10_ antioxidant activity in response to PbAc exposure, RT-qPCR was used to detect Nrf2 and HO-1, which activate cellular antioxidants and detoxifying molecules. Rats exposed to PbAc showed a significantly (*P* < 0.05) downregulated cortical Nrf2 and a significantly (*P* < 0.05) upregulated cortical HO-1 mRNA expression compared with the control rats. No significant change in cortical Nrf2 and HO-1 mRNA expression was observed in the CoQ_10_-treated rats compared with the control rats. However, CoQ_10_ treatment increased the expression of Nrf2 and decreased the expression of HO-1 compared with PbA-exposure. These findings reflect the potent antioxidant activity and protective effects of CoQ_10_ against PbAc-induced oxidative stress in cortical tissue (Figure 4).

### 3.3. PbAc-Induced Inflammation in the Cortical Tissue

To assess the anti-inflammatory activity of CoQ_10_ against PbAc, we measured the concentration of IL-1β, TNF-α, and IL-10 in cortical homogenates and their gene expressions using ELISA and qRT-PCR methods. PbAc exposure increased the release of pro-inflammatory cytokines (IL-1β and TNF-α) and their mRNA expression, while decreased the anti-inflammatory cytokine (IL-10) and its gene expression compared with the control group. No significant change in IL-1β, TNF-α, or IL-10 concentration and their gene expression were observed in the cortical tissue of CoQ_10_-injected rats compared with the control rats. Furthermore, CoQ_10_ significantly (*P* < 0.05) attenuated the inflammatory effects promoted by PbAc exposure in the cortical tissue by decreasing the production and gene expression of IL-1β and TNF-α and increasing IL-10 level and its gene expression (Figure 5).

### 3.4. PbAc-Induced Apoptotic Cascade in the Cortical Tissue

The anti-apoptotic effects of CoQ_10_ against PbAc-induced neuronal damage and apoptotic cascade activation were also investigated. Rats injected with PbAc showed upregulated cortical Bax and caspase-3 mRNA expression, and downregulated Bcl-2 mRNA expression compared with the control rats. No significant changes in the expression of these apoptotic proteins were observed in the CoQ_10_-injected rats compared with the control rats. Furthermore, CoQ_10_ treatment significantly (*P* < 0.05) altered the expression of pro- and anti-apoptotic proteins in cortical tissues compared with PbAc-exposure (Figure 6).

### 3.5. PbAc-Induced Neurochemical Alterations in the Cortical Tissue

The neuroprotective role of CoQ_10_ against PbAc-induced neurochemical alterations in cortical homogenate was elucidated. PbAc-injection significantly increased the cortical levels of 5-HT, DA, NE, GABA, and glutamate, and decreased the levels of ATP compared with the control group. No significant change in the neurochemical parameters was observed in the CoQ_10_-treated rats; however, CoQ_10_ modulated monoaminergic and aminoacidergic transmission following PbAc-injection by returning the neuromediator levels close to the control values (Table 2).

### 3.6. PbAc-Induced Histopathological Alterations in the Cortical Tissue

Neural tissue of the control and CoQ_10_ injected groups exhibited a normal histological structure. PbAc-exposed rats showed degenerated and pyknotic cortical neurons associated with the presence of inflammatory and apoptotic cells (Figure 7). Meanwhile, the treatment with CoQ_10_ alleviated markedly the cortical tissue alterations produced by PbAc; however, still some cortical neurons remained damaged.

## 4. Discussion

Exposure to heavy metals, even at low doses, has been associated with the development of severe health problems. This study explored the potential protective effects of CoQ_10_ against Pb-induced neuronal damage, and demonstrated increased Pb concentration in the cortical tissue following PbAc injection. The cerebral cortex has been recognized as a target for Pb intoxication [27] as lead can penetrate the blood brain barrier, thus its accumulation has been linked with several neurological impairments such as mental retardation, schizophrenia, Alzheimer’s disease, and Parkinsonism [28,29]. It has been suggested that antioxidants and phenolics can penetrate the blood brain barrier and chelate heavy metals [30]. In this study, the injection of CoQ_10_ following PbAc was able to reduce the Pb concentration in cortical tissue.

Numerous studies in animals and humans have confirmed the involvement of oxidative damage in lead-induced neurotoxicity. We reported the altered redox status of cortical tissue following PbAc exposure, such as elevated MDA and NO levels and decreased GSH content and SOD, CAT, GPx, and GR activity. Pb also downregulated the expression of cortical Nrf2 and HO-1, which provide cellular protection against oxidative stress. Pb is known to enhance the production of reactive oxygen species (ROS), which attack cellular compartments and promote the peroxidation of cell membrane lipids, disrupting the integrity and function of the cell membrane and resulting in cell death [31]. Pb can also contribute to mitochondrial dysfunction resulting in inhibition of mitochondrial electron transport chain at complexes II, III and IV level, depleting energy production and mitochondrial membrane potential in both in vivo and in vitro models [32]. In addition, the overexpression of iNOS, which is responsible for NO synthesis, may explain the elevated levels of cortical NO observed in this study. ROS produced following Pb exposure have been found to activate nuclear factor-κB (NF-κB), which in turn promotes the upregulation of microglial iNOS and thus NO release [33]. Furthermore, acute Pb exposure has been found to decrease the level of antioxidant molecules in the brain [13]. Pb is known to form covalent bonds with the –SH group in GSH, GPx, and GR, which eventually get depleted; GSH was found to be reduced during Pb detoxification by its secretion in bile. This process may explain the depleted cortical GSH content we observed, which may increase lipid peroxidation and the development of oxidative damage [34]. Pb can also inactivate SOD and CAT by replacing zinc ions, which act as a cofactor to regulate the activity of these antioxidants [35]. The transcription factors Nrf2 and HO-1 provide cytoprotection by activating the expression of antioxidants and activating the detoxifying system against different toxicants. In some conditions, however, the stimulation of HO-1 may cause damage and promote cytotoxicity by enhancing the products of heme degradation, such as iron ions, biliverdin and carbon monoxide [36]. The overproduction of the pro-inflammatory cytokines including TNF-α and IL-1β was found to be associated with the upregulation of HO-1 expression in the glial cells. Additionally, the hyperactivity of HO-1 in the neuronal tissue potentiates iron mitochondrial sequestration which further enhances energy depletion and iron deposition, resulting in the progression of neurodegenerative disorders [37]. Numerous studies have recorded alterations in the Nrf2/HO-1 pathway in neural tissues exposed to Pb [38,39]. The downregulated gene expression of antioxidants following acute Pb exposure in this study may explain the inactivation of these cortical antioxidants. Meanwhile, the downregulation of Nrf2 may explain the decreased gene expression of antioxidants and their inactivation in this study.

Treatment with CoQ_10_ significantly reduced the elevated level of MDA in cortical tissue following PbAc exposure. CoQ10 is an essential cofactor for the inner mitochondrial enzyme, dihydrooratate dehydrogenase which is involved in de novo pyrimidine synthesis [40]. It has been demonstrated that CoQ_10_ prevents increased lipid peroxidation in red blood cells following cadmium exposure by quenching peroxide radicals, thus CoQ_10_ deficiency is associated with increased lipid peroxidation [11]. In another experimental model, CoQ_10_ also prevented increased lipid peroxidation in the cortical tissue in response to arsenic exposure [10]. The suppression of lipid peroxidation observed in this study may be due to the ability of CoQ_10_ to scavenge ROS, particularly peroxide radicals. CoQ_10_ treatment also reduced the elevated level of cortical NO following PbAc exposure. Various experimental models have reported the ability of CoQ_10_ to attenuate excessive NO release and have suggested that CoQ_10_ suppresses iNOS expression [41,42], corresponding with the results presented here. Furthermore, CoQ_10_ was found to counteract the oxidative burst produced following Pb exposure by increasing the activity of GSH, SOD, CAT, GPx, and GR, and the cortical expression of Nrf2 and HO-1. We observed increased levels of these antioxidants, which may be due to the upregulation of their gene expression. These results concur with previous studies of antioxidant activity and the protective effect of CoQ_10_ in different tissues [10,11,43].

Neurochemical alterations were observed in this study following PbAc injection, namely the increased cortical content of 5-HT, DA, NE, glutamate, and GABA, and decreased ATP levels. Disturbed monoaminergic and aminoacidergic transmission has been associated with motor, behavioral, and cognitive impairments [44], whilst it has been suggested that Pb can replace the calcium necessary for neurotransmission, resulting in reduced neurotransmitter release [45]. Pb was found to inhibit the function of sodium ions at low doses, thus blocking the propagation of action potentials required to release neurotransmitters [46]. Elevated levels of monoamines have previously been attributed to an increase in their synthesis [47]. Similarly, Gupta and Gill [48] showed that tyrosine uptake and tyrosine hydroxylase activity were significantly increased following Pb treatment. Moreover, Pb can alter the structure and function of monoamine oxidase by binding its thiol group [49]. Pb has also been found to inhibit energy metabolism by disrupting Na^+^/K^+^ ATPase [50]. Hippocampal neurons exposed to Pb showed reduced excitatory and inhibitory postsynaptic currents, regulated by the release of neurotransmitters from presynaptic neurons, which may be due to the inhibition of GABA and glutamate release [51]. Moreover, reduced GABA and glutamate release may be due to the downregulation of synaptophysin and synaptobrevin, which are important for the vesicular release of these neurotransmitters [3].

CoQ_10_ treatment significantly improved the reduced neurotransmission and energy metabolism in cortical tissue following Pb exposure. CoQ_10_ has been demonstrated to enhance serotonin neurotransmission by activating serotonergic receptors in a chronic unpredictable mild stress rat model [52]. Furthermore, CoQ_10_ increased the concentration of 5-HT and NE in a model of diabetes mellitus, likely because CoQ_10_ can inhibit oxidative damage in neuronal tissues [53]. CoQ_10_ has also been shown to protect dopaminergic neurons against iron-induced mitochondrial dysfunction and cell death [54], whilst CoQ_10_ supplementation reduced the elevated tyrosine hydroxylase activity and NE concentration of the hypothalamic paraventricular nucleus in a hypertension model [55]. CoQ_10_ acts as an electron carrier in the electron transport chain during oxidative phosphorylation and ATP production in the mitochondria, thus CoQ_10_ reversed the decline in oxidative phosphorylation efficiency in aged diabetic rats and enhanced ATP production [56]. In agreement with our findings, Duberley et al. [40] demonstrated that CoQ_10_ supplementation restored neuronal mitochondrial electron transport chain enzyme activities. However, the authors reported that to achieve the neurological effect of CoQ_10_, the dose of CoQ_10_ should be higher than the current oral CoQ_10_ formulations due to the poor transfer of CoQ_10_ across the blood-brain barrier, or an inability of CoQ_10_ deficient neurons to utilize exogenous CoQ_10_.

Neuroinflammation is one of the fundamental mechanisms of Pb-induced neurotoxicity. PbAc treatment triggered an inflammatory response associated with the excessive production of IL-1β and TNF-α and reduced IL-10 in cortical tissue. This was accompanied with overexpression of IL-1β and TNF-α and downregulation of IL-10 genes expression. Several studies have revealed that IL-1β and TNF-α were overexpressed in cortical tissue following Pb exposure. These chemical messengers are upregulated in response to Pb exposure due to the activation of NF-κB signaling, which promotes the release of pro-inflammatory cytokines [57,58]. The increased levels of these inflammatory mediators may also be due to their upregulated gene expression in the case of IL-1β and TNF-α, and the downregulation of IL-10 gene expression which we reported in this study. The concentration of IL-10 is decreased in various regions of the brain in response to Pb exposure, including the cerebral cortex [59]. Treatment with CoQ_10_ inhibited the Pb-induced inflammatory response by reducing the elevated cortical levels of IL-1β and TNF-α and downregulating their gene expression. It has been demonstrated that CoQ_10_ can exert anti-inflammatory effects by inactivating NF-κB, which is responsible for the overexpression of pro-inflammatory cytokines [60]. CoQ10 was found to enhance IL-10 production in a menopausal rat model following spinal cord damage [61]. Elevated IL-10 levels can reportedly block the expression of IL-1, which has a pro-inflammatory role in brain tissue [62].

Heavy metal exposure has been associated with neuronal injury via apoptosis, a hypothesis that was confirmed in this study. PbAc-injected rats showed the upregulation of cortical pro-apoptotic proteins, including Bax and caspase-3, and the downregulation of the anti-apoptotic protein Bcl-2. Several studies have demonstrated the upregulation of Bax and caspase-3 and the downregulation of Bcl-2 in brain tissues following Pb exposure. The activation of the apoptotic cascade in response to Pb exposure may be due to ROS overproduction [63,64]. The downregulation of the Nrf2/HO-1 pathway following Pb injection observed in this study may also explain the cortical upregulation of Bax and caspase-3, and the downregulation of Bcl-2. CoQ10 also modulated the expression of these cortical proteins and has been reported to upregulate anti-apoptotic mitochondrial uncoupling proteins via its antioxidant activity [65]. In addition, CoQ10 improved cerebral infarction and exerted anti-apoptotic effects by modulating apoptotic proteins in rats with stroke [66]. The anti-apoptotic effects of CoQ10 against iron-induced apoptosis in dopaminergic neurons have also been shown [54]. Consequently, based on the obtained findings, Figure 8 schematically illustrates the potential neuroprotective effect of CoQ10 against lead acetate-induced neurotoxicity.

## 5. Conclusions

In this study, CoQ_10_ administration protected against Pb-induced cortical damage by maintaining the balance between oxidants and antioxidants, and enhancing Nrf2/HO-1 expression. It also showed anti-inflammatory activity by reversing elevated IL-1β and TNF-α levels, anti-apoptotic activity by inhibiting pro-apoptotic and enhancing anti-apoptotic protein expression, and modulated neurotransmission and energy metabolism.

## Figures and Tables

**Figure 1 ijerph-16-02895-f001:**
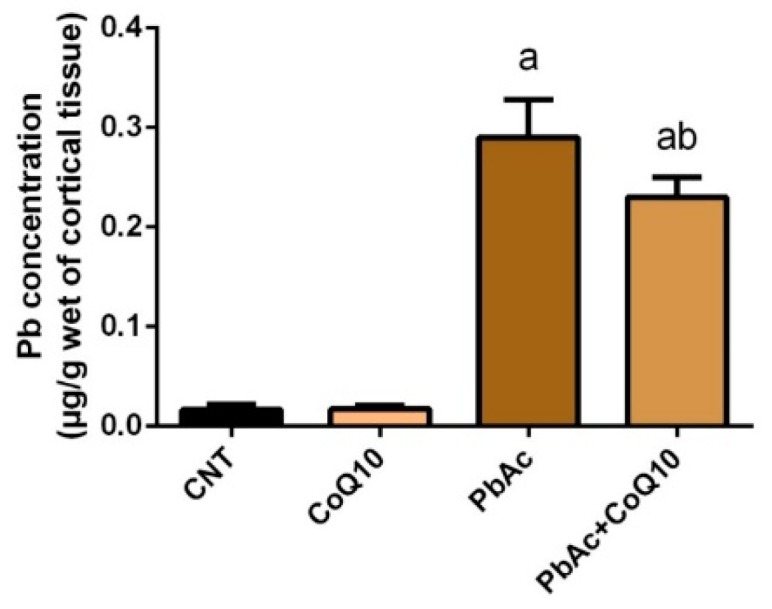
The effects of lead acetate and/or coenzyme Q10 on cortical lead concentration. Results represent the mean ± SD (*n* = 7); significant changes at *P* < 0.05 were determined using Duncan’s *post-hoc* test. ^a^ significant change compared with the control group; ^b^ significant change compared with the PbAc injected group.

**Figure 2 ijerph-16-02895-f002:**
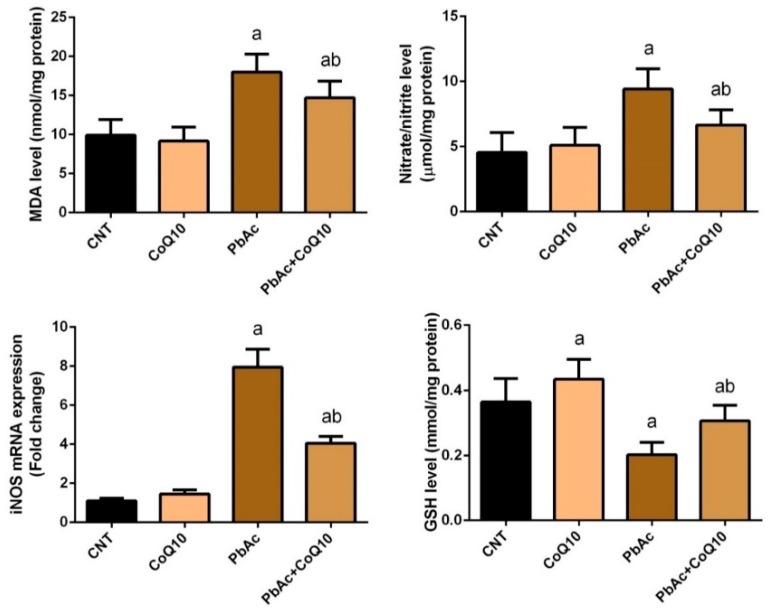
The beneficial effects of coenzyme Q10 on cortical lipid peroxidation and nitrate/nitrite and glutathione content following lead acetate exposure. Results represent the mean ± SD (*n* = 7); significant changes at *P* < 0.05 were determined using Duncan’s *post-hoc* test. ^a^ significant change compared with the control group; ^b^ significant change compared with the PbAc injected group.

**Figure 3 ijerph-16-02895-f003:**
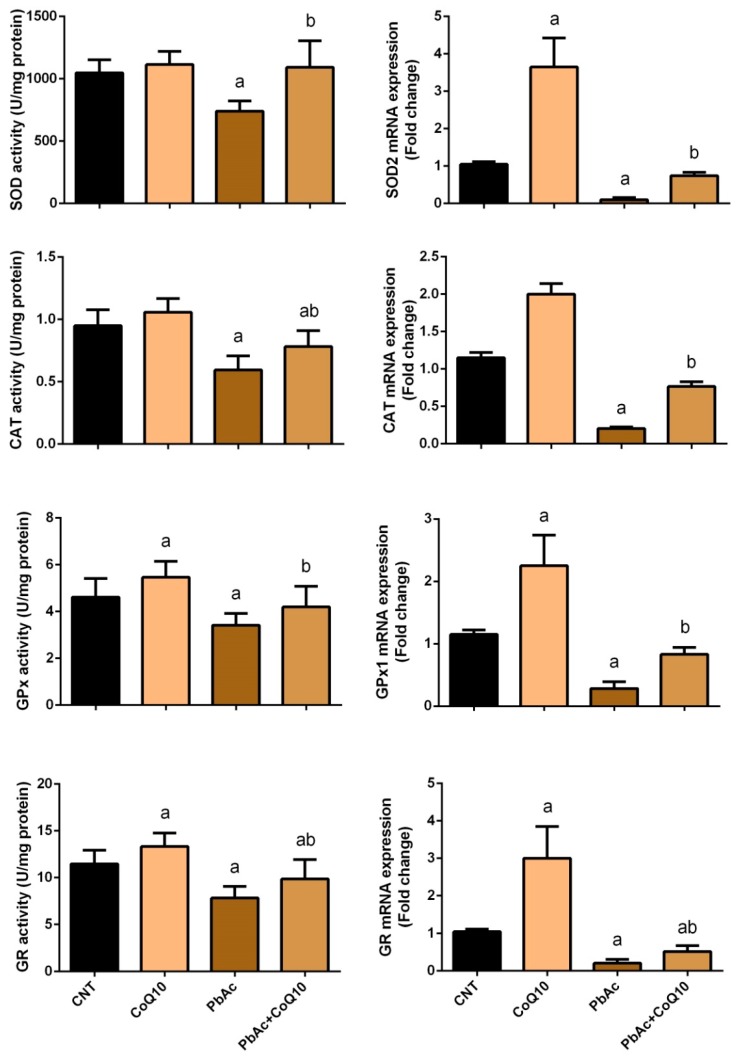
Effects of coenzyme Q10 on cortical SOD, CAT, GPx, and GR activity and their mRNA expression following lead acetate exposure. Results represent the mean ± SD (*n* = 7); significant changes at *P* < 0.05 were determined using Duncan’s post-hoc test. ^a^ significant changes compared with the control group; ^b^ significant changes compared with the PbAc injected group. mRNA expression data are expressed as the mean ± SD of triplicate assays normalized to β-actin and expressed as fold changes (log2 scale) compared with the mRNA levels of the control group.

**Figure 4 ijerph-16-02895-f004:**
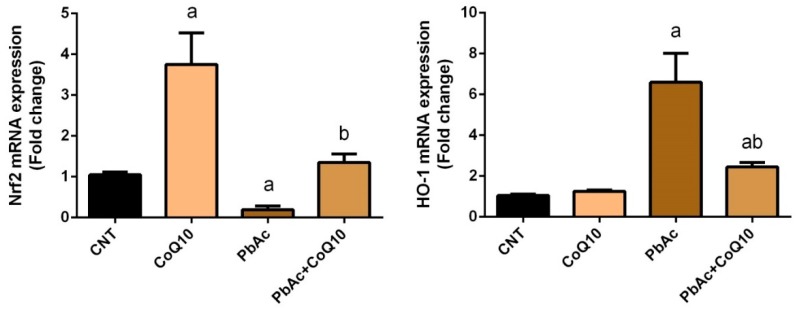
Effects of coenzyme Q10 on cortical Nrf2 and HO-1 mRNA expression following lead acetate exposure. mRNA Data are expressed as the mean ± SD of triplicate assays normalized to β-actin and expressed as fold changes (log2 scale) compared with the mRNA levels of the control group; significant changes at *P* < 0.05 were determined using Duncan’s post-hoc test. ^a^ significant changes compared with the control group; ^b^ significant changes compared with the PbAc injected group.

**Figure 5 ijerph-16-02895-f005:**
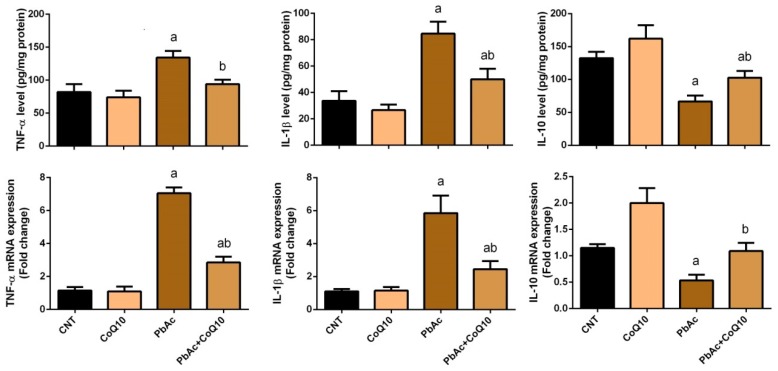
Effects of coenzyme Q10 on cortical levels and expression of IL-1β, TNF-α, and IL-10 following lead acetate exposure. Results of ELISA were represented as the mean ± SD (*n* = 7); significant changes at *P* < 0.05 were determined using Duncan’s *post-hoc* test. ^a^ significant changes compared with the control group; ^b^ significant changes compared with the PbAc injected group. mRNA Data are expressed as the mean ± SD of triplicate assays normalized to β-actin and expressed as fold changes (log2 scale) compared with the mRNA levels of the control group.

**Figure 6 ijerph-16-02895-f006:**
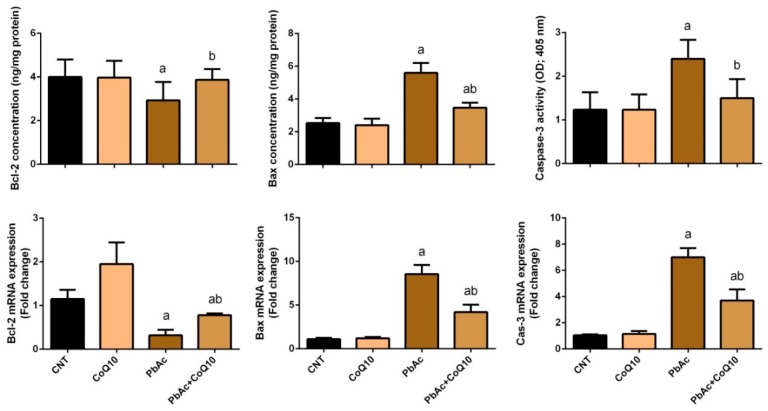
Effects of coenzyme Q10 on the cortical levels and gene expression of Bax, caspase-3, and Bcl-2 in mouse cortex neurons following lead acetate exposure. mRNA Results are represented as the mean ± SD (n = 7); significant changes at *P* < 0.05 were determined using Duncan’s post-hoc test. ^a^ significant changes compared with the control group; ^b^ significant changes compared with the PbAc injected group. Data for qRT-PCR are expressed as the mean ± SD of triplicate assays normalized to β-actin and expressed as fold changes (log2 scale) compared with the mRNA levels of the control group.

**Figure 7 ijerph-16-02895-f007:**
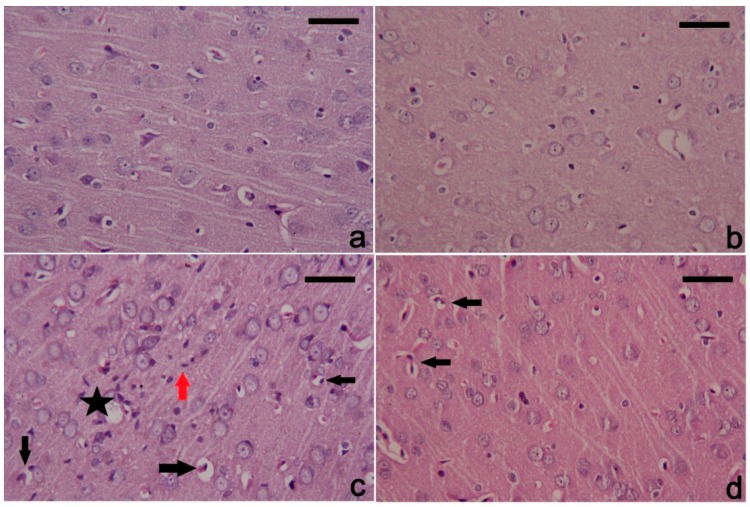
Histopathological changes in cortical tissue following treatment with coenzyme Q10 and lead acetate. (**a**) and (**b**) showing normal cortical structure of control and coenzyme Q10 treated rats; (**c**) section from the lead acetate treated rat showing intracellular and extracellular vacuoles and oedema around the cell (black star), many inflammatory cell infiltration (red arrow), and degenerated neurons with deeply stained nuclei indicating apoptosis (black arrow). (**d**) post-treatment with coenzyme Q10 markedly attenuated all cortical damage caused by lead acetate. However, a few degenerative neurons in the cortical tissue are still found. Hematoxylin and eosin (H&E). Scale bar 100 nm.

**Figure 8 ijerph-16-02895-f008:**
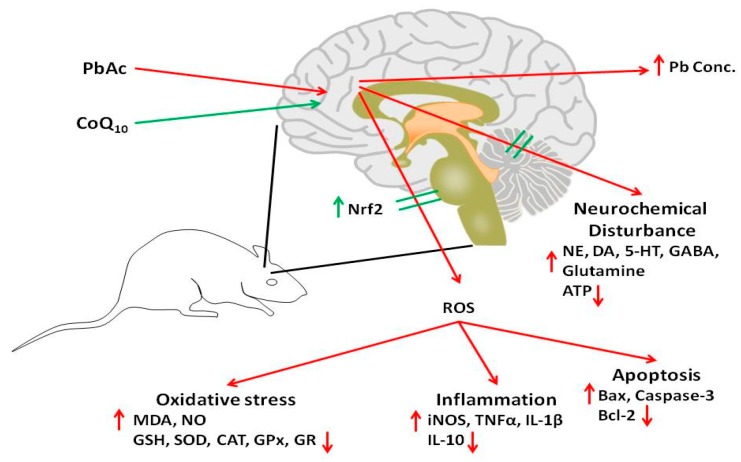
A summarized figure describes the potential protective role of CoQ10 against lead acetate-induced neurotoxicity; CoQ10: coenzyme Q10; PbAc: lead acetate; Nrf2: nuclear factor erythroid 2–related factor 2; ROS: reactive oxygen species; MDA: malondialdehyde; NO: nitric oxide; GSH: glutathione; SOD: superoxide dismutase; CAT: catalase; GPx: glutathione peroxidase; GR: glutathione reductase; iNOS: inducible nitric oxide dismutase; TNF-α: tumor necrosis factor-alpha; 1L-1β: interleukin-1 beta; 1L-10: interleukin-10; Bax: BCL2-associated X protein; Bcl-2: B-cell lymphoma; NE: norepinephrine; DA: dopamine; 5-HT: serotonin.

**Table 1 ijerph-16-02895-t001:** Details of genes symbol, accession number, primer sequences, amplicon size and PCR efficiency (%)* analyzed in Real-Time PCR.

Name	Gene Symbol and Accession No.	Sense (5’---3’)	Antisense (5’---3’)	Amplicon Size (bp)	PCR Efficiency (%)
**β-actin**	*Actb*: NM_031144.3	GGCATCCTGACCCTGAAGTA	GGGGTGTTGAAGGTCTCAAA	203	96.9%
**SOD2**	*Sod2*: NM_017051.2	AGCTGCACCACAGCAAGCAC	TCCACCACCCTTAGGGCTCA	191	106.8%
**CAT**	*Cat*: NM_012520.2	TCCGGGATCTTTTTAACGCCATTG	TCGAGCACGGTAGGGACAGTTCAC	362	93.7%
**GPx1**	*Gpx1*: NM_030826.4	CGGTTTCCCGTGCAATCAGT	ACACCGGGGACCAAATGATG	245	103.2%
**GR**	*Gsr*: NM_053906.2	TGGCACTTGCGTGAATGTTG	CGAATGTTGCATAGCCGTGG	233	116.0%
**Nrf2**	*Nfe2l2*: NM_031789.2	GGTTGCCCACATTCCCAAAC	GGCTGGGAATATCCAGGGC	116	105.5%
**HO-1**	*Hmox1*: NM_012580.2	GCGAAACAAGCAGAACCCA	GCTCAGGATGAGTACCTCCCA	185	105.2%
**iNOS**	*Nos2*: NM_012611.3	GTTCCTCAGGCTTGGGTCTT	TGGGGGAACACAGTAATGGC	825	97.7%
**TNF-α**	*Tnfa*: NM_012675.3	AGAACTCAGCGAGGACACCAA	GCTTGGTGGTTTGCTACGAC	461	104.7%
**IL-1β**	*Il1b*: NM_031512.2	GACTTCACCATGGAACCCGT	GGAGACTGCCCATTCTCGAC	104	109.3%
**IL-10**	*Il10*: NM_012854.2	TTGAACCACCCGGCATCTAC	CCAAGGAGTTGCTCCCGTTA	91	106.8%
**Bcl-2**	*Bcl2*: NM_016993.1	ACTCTTCAGGGATGGGGTGA	TGACATCTCCCTGTTGACGC	94	96.7%
**Bax**	*Bax*: NM_017059.2	CTGAGCTGACCTTGGAGC	GACTCCAGCCACAAAGATG	413	102.7%
**Caspase-3**	*Casp3*: NM_012922.2	GAGCTTGGAACGCGAAGAAA	TAACCGGGTGCGGTAGAGTA	635	108.4%

*: PCR efficiency (%) was attached as a Appendix A.

**Table 2 ijerph-16-02895-t002:** Effect of coenzyme Q10 on cortical norepinephrine, dopamine, serotonin, glutamate, GABA and ATP content following lead acetate exposure.

Brain Areas	Experimental Groups			
	CNT	CoQ10	PbAc	PbAc + CoQ10
**Norepinephrine (μg/ g tissue)**	0.41 ± 0.003	0.39 ± 0.002	0.88 ± 0.002^a^	0.62 ± 0.002^ab^
**Dopamine (μg/g tissue)**	1.07 ± 0.007	1.14 ± 0.04	1.76 ± 0.03^a^	1.38 ± 0.001^b^
**Serotonin (μg/g tissue)**	0.48 ± 0.002	0.58 ± 0.002	0.93 ± 0.02^a^	0.66 ± 0.002^b^
**glutamate (μmol/g tissue)**	9.48 ± 0.06	9.17 ± 0.03^a^	12.88 ± 0.03^a^	10.72 ± 0.04^b^
**GABA (μmol/g tissue)**	4.39 ± 0.02	4.22 ± 0.03	7.39 ± 0.04^a^	5.39 ± 0.03^b^
**ATP (μmol/g tissue)**	5.03 ± 0.03	7.93 ± 0.03^a^	2.39 ± 0.03	6.48 ± 0.04^b^

Results represent the mean ± SD (*n* = 7); significant changes at *P* < 0.05 were determined using Duncan’s post-hoc test. ^a^ significant changes compared with the control group; ^b^ significant changes compared with the PbAc injected group.

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
