# Peer review of "The Neuroprotective Role of Coenzyme Q10 Against Lead Acetate-Induced Neurotoxicity Is Mediated by Antioxidant, Anti-Inflammatory and Anti-Apoptotic Activities"

_ijerph, 2019, doi:10.3390/ijerph16162895_

Round 1

Reviewer 1 Report

The paper describes the effects of coenzyme Q10 treatment after exposure to very high concentrations of lead on neuronal tissues in mice.  I have several serious concerns with this manuscript that would need to be addressed before I could recommend it for acceptance (see below points in bold).

Major issues

- Introduction

this section lacks any real justification of why this research is important and timely.  You need to really highlight why this work is important, timely and novel.

- Methods

This section lacks appropriate level of detail.  See the comments below for specific detail.

section 2.3.  You need to justify the chosen dose of lead.  For example, does this dose reflect the pharmacokinetics of lead in humans?  Why was a 5 day repeated exposure selected?  The study is fundamentally compromised if the dosing either does not induce a similar physiological response as seen in humans, or does not appropriately mimic physiologically-relevant exposures.

section 2.5.  The hydrogen peroxide assay is not a widely reported one (the reagents used are often used for the assessment of peroxidase activity and uric acid metabolism (using a detection system involving hydrogen peroxide production).  Thus a more detailed explanation of the assay (or suitable references) are required to allow readers to understand how it works and the implications of this on its specificity for hydrogen peroxide detection.  If this is something developed and tested in house then the validation data should be referenced or presented.

section 2.5. for all the assays in this section you need to highlight key information relating to calibration curve ranges, incubation times etc. 

section 2.6.  Similar to the above comments, the information given about these assays is minimal.  You need sufficient detail to explain how the assays work and to allow readers to repeat your methods.

Section 2.7.  You need to say which kits you used (Manufacturer and kit name at the very least).

section 2.8. qPCR information is too brief and vague.  You should report your method in line with the MIQE guidelines for qPCR publication information.

Line 143.  Table 1 is missing the gene accession numbers for the genes (it is stated in the text on lines 141-142 that these were included in the table).

section 2.11. You need to say which kits you used (manufacturer and kit name at the very least). 

Section 2.13.  This section lacks detail.  Did you test your data for normality of distribution and homogeneity of variance?  What type of ANOVA did you use (repeated measures or independent measures)? What software was used to do this? 

section 2.13 and figure legends.  The text section says error bars are SEM and the legends say SD.  Which is correct?

Results

I find the statistical significances unconvincing based on the plotted graphs.  Many of the reported significances show data plotted with overlapping error bars for compared groups - suggesting that these datasets have significant overlap.  This is further complicated by the error bars potentially being SEM rather than SD.  I am also not fully convinced the data is normally distributed (hence the need for normality testing).  If the data is not normally distributed you should plot median values +/- interquartile ranges (a quick way to assess this is to compare the mean and median values for a dataset - if the values are the same/very similar it is likely to be normally distributed).

Figure 2 - should the nitrate/nitrite graph be labelled nitric oxide (y-axis label should reflect this)

Figure 3 - GR activity units need checking and amending.

Line 244.  You state that you want to look at Pb-induced neuronal loss, however you do not appear to have measured this loss at all, so how can you say that it happens?

Table 2.  what is plotted in this table (mean +/- SD?  SEM?)

Figure 7.  The figure legend says cadmium rather than lead.  This needs changing

Discussion

This section may well need re-writing in light of any statistical analysis changes and data re-interpretation.

Line 335.  States ATP levels decreased, which is not what is shown in table 2.  Which is correct?

Author Response

We very much appreciated the comments and suggestions made by the reviewer. We complied with these comments in our revised manuscript. The changes we made using track change tool.

Reviewer 1:

The paper describes the effects of coenzyme Q10 treatment after exposure to very high concentrations of lead on neuronal tissues in mice.  I have several serious concerns with this manuscript that would need to be addressed before I could recommend it for acceptance (see below points in bold).

Major issues

- Introduction

this section lacks any real justification of why this research is important and timely.  You need to really highlight why this work is important, timely and novel.

Response: Thanks for your comment; we have followed your suggestion and we have covered these points in the introduction.

- Methods

This section lacks appropriate level of detail.  See the comments below for specific detail.

Section 2.3.  You need to justify the chosen dose of lead.  For example, does this dose reflect the pharmacokinetics of lead in humans?  Why was a 5 day repeated exposure selected?  The study is fundamentally compromised if the dosing either does not induce a similar physiological response as seen in humans, or does not appropriately mimic physiologically-relevant exposures.

Response: Thanks for your comment; this just a preclinical study and the used dose represents 1/30 of the LD50 of lead acetate for rats and the sentence was modified accordingly.

Section 2.5.  The hydrogen peroxide assay is not a widely reported one (the reagents used are often used for the assessment of peroxidase activity and uric acid metabolism (using a detection system involving hydrogen peroxide production).  Thus a more detailed explanation of the assay (or suitable references) are required to allow readers to understand how it works and the implications of this on its specificity for hydrogen peroxide detection.  If this is something developed and tested in house then the validation data should be referenced or presented.

Response: Thanks for your comment; we are sorry for this mistake and we have deleted the mentioned method.

Section 2.5. for all the assays in this section you need to highlight key information relating to calibration curve ranges, incubation times etc. 

Response: Thanks for your comment; we have followed your suggestion and more details were added.

section 2.6.  Similar to the above comments, the information given about these assays is minimal.  You need sufficient detail to explain how the assays work and to allow readers to repeat your methods.

Response: Thanks for your comment; we have followed your suggestion and more details were added.

Section 2.7.  You need to say which kits you used (Manufacturer and kit name at the very least).

Response: Thanks for your comment; we have followed your suggestion

section 2.8. qPCR information is too brief and vague.  You should report your method in line with the MIQE guidelines for qPCR publication information.

Response: Thanks for your comment; we have followed your suggestion.

Line 143.  Table 1 is missing the gene accession numbers for the genes (it is stated in the text on lines 141-142 that these were included in the table).

Response: Thanks for your comment; we are sorry for the mistake and this section was modified.

section 2.11. You need to say which kits you used (manufacturer and kit name at the very least). 

Response: Thanks for your comment; we have followed your suggestion

Section 2.13.  This section lacks detail.  Did you test your data for normality of distribution and homogeneity of variance?  What type of ANOVA did you use (repeated measures or independent measures)? What software was used to do this? 

Response: Thanks for your comment; we have tested the data using normal distribution test and we used parametric analysis. In the current study we employed One-way ANOVA, then we used Tukey’s post hoc test to compare between the differences between different groups using SPSS software.

section 2.13 and figure legends.  The text section says error bars are SEM and the legends say SD.  Which is correct?

Response: Thanks for your comment; we have corrected this mistake throughout the manuscript.

Results

I find the statistical significances unconvincing based on the plotted graphs.  Many of the reported significances show data plotted with overlapping error bars for compared groups - suggesting that these datasets have significant overlap.  This is further complicated by the error bars potentially being SEM rather than SD.  I am also not fully convinced the data is normally distributed (hence the need for normality testing).  If the data is not normally distributed you should plot median values +/- interquartile ranges (a quick way to assess this is to compare the mean and median values for a dataset - if the values are the same/very similar it is likely to be normally distributed). 

Response: Thanks for your comment; we are sorry for this mistake. We have used SD not SEM and we corrected this mistake.

Figure 2 - should the nitrate/nitrite graph be labelled nitric oxide (y-axis label should reflect this)

Response: Thanks for your comment; we have mentioned in the result section that nitrate/nitrite refers to the nitric oxide level.

Figure 3 - GR activity units need checking and amending.

Response: Thanks for your comment; we have corrected this mistake.

Line 244.  You state that you want to look at Pb-induced neuronal loss, however you do not appear to have measured this loss at all, so how can you say that it happens?

Response: Thanks for your comment; this statement was modified to avoid any confusion.

Table 2.  what is plotted in this table (mean +/- SD?  SEM?)

Response: Thanks for your comment; we have unified the used method.

Figure 7.  The figure legend says cadmium rather than lead.  This needs changing

Response: Thanks for your comment; we have corrected this mistake.

Discussion

This section may well need re-writing in light of any statistical analysis changes and data re-interpretation.

Response: Thanks for your comment; we didn’t change the discussion as our statistical analysis is correct as we think.

Line 335.  States ATP levels decreased, which is not what is shown in table 2.  Which is correct?

Response: Thanks for your comment; we are sorry for this mistake. The used ATP data was wrong in the PbAc and CoQ10 treated groups and we correct it.

Once again we thank the reviewer for his valuable comments and suggestions, which helped us to improve our manuscript. We hope that you can accept our revised paper as it stands now.

Best regards,

Prof.Dr. Manal Elkhadragy,

Zoology Department, Faculty of Science, King Saud University, Riyadh 11451, Saudi Arabia

Reviewer 2 Report

General comments

The Authors reported that Wistar albino rats were divided into 4 equal groups (n = 7) and treated as follows: control group injected with physiological saline (0.9% NaCl); CoQ10 group injected with CoQ10 (10 mg/kg); PbAc group injected with PbAc (20 mg/kg); PbAc+CoQ10 group injected first with PbAc and after 1 h, with CoQ10. The obtained results showed that PbAc significantly increased cortical lipid peroxidation, nitrate/nitrite levels, and inducible nitric oxide synthase expression and decreased glutathione content, superoxide dismutase, catalase, glutathione peroxidase, glutathione reductase activity and mRNA expression.

Furthermore, treatment with CoQ10 rescued cortical neurons from PbAc-induced neurotoxicity by restoring the balance between oxidants and antioxidants, activating the Nrf2/HO-1 pathway, suppressing inflammation, inhibiting the apoptotic cascade, and modulating cortical neurotransmission and energy metabolism.

Overall, the paper is clear and scientifically sound, and the reported results are of potential interest; Notwithstanding, I would appreciate if the Authors carefully consider and address the issues below.

Specific Comments

- My major concern about this interesting paper is that nevertheless the title of the paper states “The Neuroprotective Role of Coenzyme Q10 Against 2 Lead Acetate-Induced Neurotoxicity Is Mediated by the Activation of the Nrf2/HO-1 Pathway”, the result and discussion sections are focused on the antioxidant and anti-inflammatory actions of the CoQ10 in PbAc treated rats, while the Authors investigated the Nrf2/HO-1 pathway only in little depth.

- In more details, at page 5 line 199 the Authors reported: “Rats exposed to PbAc showed a significantly (P < 0.05) 199 downregulated cortical Nrf2 and HO-1 mRNA expression compared with the control rats. No significant change in cortical Nrf2 and HO-1 mRNA expression was observed in the CoQ10-treated rats compared with the control rats. However, CoQ10 treatment increased the expression of Nrf2 and HO-1 compared with PbA-exposure. These findings reflect the potent antioxidant activity and protective effects of CoQ10 against PbAc-induced oxidative stress in cortical tissue”. To some data interpretations are not clear, in figure 4, it is possible to understand that PbAc-treatment down-regulates Nrf2 but up-regulates HO-1. Can the Authors clarify this point? Furthermore, in this experimental setting, it seems that HO-1 expression is not linked to Nrf2 activation; for example, CoQ10 upregulates Nrf2 but not HO-1. It should be interesting to evaluate if Nrf2 protein is translocated into the nucleus in response to CoQ10-treatment. Moreover, how the Authors explain the up-regulation of HO-1 in response to PbAc-exposure along with the downregulation of Nrf2 mRNA. Can other signal pathways be involved in the HO-1 mRNA transcription?

Minor Points:

- Page 11 line 341: “Similarly, [40] showed that tyrosine uptake and”. please verify the format of the cited reference.

Author Response

We very much appreciated the comments and suggestions made by the reviewer. We complied with these comments in our revised manuscript. The changes we made using track change tool.

Reviewer 2:

Specific Comments

- My major concern about this interesting paper is that nevertheless the title of the paper states “The Neuroprotective Role of Coenzyme Q10 Against 2 Lead Acetate-Induced Neurotoxicity Is Mediated by the Activation of the Nrf2/HO-1 Pathway”, the result and discussion sections are focused on the antioxidant and anti-inflammatory actions of the CoQ10 in PbAc treated rats, while the Authors investigated the Nrf2/HO-1 pathway only in little depth.

Response: Thanks for your comment; we have followed your suggestion and the title has been modified.

- In more details, at page 5 line 199 the Authors reported: “Rats exposed to PbAc showed a significantly (P < 0.05) 199 downregulated cortical Nrf2 and HO-1 mRNA expression compared with the control rats. No significant change in cortical Nrf2 and HO-1 mRNA expression was observed in the CoQ10-treated rats compared with the control rats. However, CoQ10 treatment increased the expression of Nrf2 and HO-1 compared with PbA-exposure. These findings reflect the potent antioxidant activity and protective effects of CoQ10 against PbAc-induced oxidative stress in cortical tissue”. To some data interpretations are not clear, in figure 4, it is possible to understand that PbAc-treatment down-regulates Nrf2 but up-regulates HO-1. Can the Authors clarify this point? Furthermore, in this experimental setting, it seems that HO-1 expression is not linked to Nrf2 activation; for example, CoQ10 upregulates Nrf2 but not HO-1. It should be interesting to evaluate if Nrf2 protein is translocated into the nucleus in response to CoQ10-treatment. Moreover, how the Authors explain the up-regulation of HO-1 in response to PbAc-exposure along with the downregulation of Nrf2 mRNA. Can other signal pathways be involved in the HO-1 mRNA transcription?

Response: Thanks for your comment; we are sorry for this mistake and we correct the results and we have modified the discussion.

Minor Points:

- Page 11 line 341: “Similarly, [40] showed that tyrosine uptake and”. please verify the format of the cited reference.,.5t57

Response: Thanks for your comment; we have followed your suggestion and the reference has been modified.

Reviewer 3 Report

This is an interesting study which assesses the  neuroprotective effect of coenzyme Q10 against lead acetate-induced neurotoxicity in brain of rats.

I have some comments that I would like the authors to address:

In the abstract there should be some introductory sentence to explain the relevance of this study rather than going virtually straight into the methodology.

 It would be good to provide some more explanation for the dose of lead acetate used in this study as it is quite a high dose and would it equate to lead poisoning in humans. Although a reference is provided, it requires more detail to justify the dosage.

Were any experiments undertaken to ensure that lead acetate or CoQ10 wouldn`t interfere with the experimental assays?

The method used to determine reduced glutathione (GSH) levels is not specific for GSH and will measure all protein thiols. A sentence is required to state the limitations of the assay.

Why weren`t CoQ10 levels determined in the rat?  Is it CoQ10 or CoQ9 which is have the beneficial effects or a mixture of the two?  What levels of CoQ10 are required to be of benefit ?  What therapeutic levels of CoQ10 should be aimed for in humans when treating lead poisoning ?  Some sentences should be provided to state the limitations of this study inn terms of CoQ10 assessment.

Does lead impair mitochondrial respiratory chain function as this is a major source of ROS generation and may impact upon the results of the study? Some detail should be provided.

There is a relevant study by Duberley et al (Int J Biochem Cell Biol. 2014 May;50:60-3) which would be appropriate to include in terms of the therapeutic efficacy of CoQ10 ameliorating neuronal oxidative stress.

A schematic diagram to outline both the toxicity of lead at the neuronal cell level and therapeutic effect of CoQ10 would be good to help summarise the findings of this study

Author Response

We very much appreciated the comments and suggestions made by the reviewer. We complied with these comments in our revised manuscript. The changes we made using track change tool.

Reviewer 3:

This is an interesting study which assesses the neuroprotective effect of coenzyme Q10 against lead acetate-induced neurotoxicity in brain of rats.

I have some comments that I would like the authors to address:

In the abstract there should be some introductory sentence to explain the relevance of this study rather than going virtually straight into the methodology.

Response: Thanks for your comment; we have followed your suggestion

It would be good to provide some more explanation for the dose of lead acetate used in this study as it is quite a high dose and would it equate to lead poisoning in humans. Although a reference is provided, it requires more detail to justify the dosage.

Response: Thanks for your comment; the used dose represents 1/30 of the LD50 of lead acetate for rats and the sentence was modified accordingly.

Were any experiments undertaken to ensure that lead acetate or CoQ10 wouldn`t interfere with the experimental assays?

Response: Thanks for your comment; we have followed the protocol of the used assays and we think that lead or CoQ10 will not interfere these experimental assays.

The method used to determine reduced glutathione (GSH) levels is not specific for GSH and will measure all protein thiols. A sentence is required to state the limitations of the assay.

Response: Thanks for your comment; the method has been modified to avoid any confusion between the methods used to measure GSH and protein thiols.

Why weren`t CoQ10 levels determined in the rat?  Is it CoQ10 or CoQ9 which is have the beneficial effects or a mixture of the two?  What levels of CoQ10 are required to be of benefit ?  What therapeutic levels of CoQ10 should be aimed for in humans when treating lead poisoning ?  Some sentences should be provided to state the limitations of this study in terms of CoQ10 assessment.

Response: Thanks for your comment; actually we don’t have the facilities to determine the level of CoQ10 in the brain tissue. Moreover, we have just evaluated the neuroprotective role CoQ10. Based on our experiment, the chosen dose was found to provide protection against lead acetate-induced neurotoxicity. Additionally, we have added limitations to our study in the discussion sections.

Does lead impair mitochondrial respiratory chain function as this is a major source of ROS generation and may impact upon the results of the study? Some detail should be provided.

Response: Thanks for your comment; we have followed your suggestion and we have added more information in the discussion section.

There is a relevant study by Duberley et al (Int J Biochem Cell Biol. 2014 May;50:60-3) which would be appropriate to include in terms of the therapeutic efficacy of CoQ10 ameliorating neuronal oxidative stress.

Response: Thanks for your comment; we have added the reference to meet your suggestion.

A schematic diagram to outline both the toxicity of lead at the neuronal cell level and therapeutic effect of CoQ10 would be good to help summarise the findings of this study

Response: Thanks for your comment; we have followed your suggestion and schematic diagram describes the potential neuroprotective effect of CoQ10 against lead neurotoxicity.

Round 2

Reviewer 1 Report

The authors have addresed a number of the points raised, however there are still a couple of key issues that I feel really need to be addressed before I could recommend this article be accepted.

There needs to be some justification of the dose of lead acetate used - the authors have stated that this is 1/30 of the LD50, however this does not really justify its choice.  Why was this dose selected?  How was it chosen?  Is it based on any pharmacokinetic information from patients with neurotoxicity?

The PCR section of the methods needs to be more detailed.  It should follow the level of detail listed in the MIQE guidlines.  In particular detailed information about the amplification efficiencies of the different primer sets used, and how the confirmation that the reference gene was stable between treatments is needed in this section to provide confidence in the PCR data.

Author Response

Dear Reviewer, thanks for your comments and we complied with it as following:

#1: There needs to be some justification of the dose of lead acetate used - the authors have stated that this is 1/30 of the LD50, however this does not really justify its choice. Why was this dose selected? How was it chosen? Is it based on any pharmacokinetic information from patients with neurotoxicity?

Response: The selection of lead acetate dose was based on our previous studies, as well as literature data [13-15], which is equal to 1/30 LD50 for rats to induce acute neurotoxicity model [13].

References:

13. Abdel Moneim, A.E. Flaxseed oil as a neuroprotective agent on lead acetate-induced monoamineric alterations and neurotoxicity in rats. Biological trace element research 2012, 148, 363-370, doi:10.1007/s12011-012-9370-4.

14. Abdel Moneim, A.E.; Dkhil, M.A.; Al-Quraishy, S. Effects of Flaxseed Oil on Lead Acetate-Induced Neurotoxicity in Rats. Biol Trace Elem Res 2011, doi:10.1007/s12011-011-9055-4.

15. Ommati, M.M.; Jamshidzadeh, A.; Heidari, R.; Sun, Z.; Zamiri, M.J.; Khodaei, F.; Mousapour, S.; Ahmadi, F.; Javanmard, N.; Shirazi Yeganeh, B. Carnosine and Histidine Supplementation Blunt Lead-Induced Reproductive Toxicity through Antioxidative and Mitochondria-Dependent Mechanisms. Biol Trace Elem Res 2019, 187, 151-162, doi:10.1007/s12011-018-1358-2

#2: The PCR section of the methods needs to be more detailed. It should follow the level of detail listed in the MIQE guidlines. In particular detailed information about the amplification efficiencies of the different primer sets used, and how the confirmation that the reference gene was stable between treatments is needed in this section to provide confidence in the PCR data.

Response: We followed your comment and more details were added.